# Formation of Autoimmune Lesions Is Independent of Antibiotic Treatment in NOD Mice

**DOI:** 10.3390/ijms22063239

**Published:** 2021-03-22

**Authors:** Mami Sato, Rieko Arakaki, Hiroaki Tawara, Takaaki Tsunematsu, Naozumi Ishimaru

**Affiliations:** Department of Oral Molecular Pathology, Graduate School of Biomedical Sciences, Tokushima University, Tokushima 7708504, Japan; c301951007@tokushima-u.ac.jp (M.S.); arakaki.r@tokushima-u.ac.jp (R.A.); c302051010@tokushima-u.ac.jp (H.T.); tsunematsu@tokushima-u.ac.jp (T.T.)

**Keywords:** NOD mice, autoimmune lesion, Sjögren’s syndrome, type 1 diabetes, antibiotic treatment

## Abstract

The relationship between autoimmunity and changes in intestinal microbiota is not yet fully understood. In this study, the role of intestinal microbiota in the onset and progression of autoimmune lesions in non-obese diabetic (NOD) mice was evaluated by administering antibiotics to alter their intestinal microenvironment. Flow cytometric analysis of spleen cells showed that antibiotic administration did not change the proportion or number of T and B cells in NOD mice, and pathological analysis demonstrated that autoimmune lesions in the salivary glands and in the pancreas were also not affected by antibiotic administration. These results suggest that the onset and progression of autoimmunity may be independent of enteral microbiota changes. Our findings may be useful for determining the appropriate use of antibiotics in patients with autoimmune diseases who are prescribed drugs to maintain systemic immune function.

## 1. Introduction

Autoimmune disease is a multifactorial disorder, and its pathogenesis is a complicated process that involves the immune system, target organs, and the internal environment [1,2,3]. Intestinal microbiota play a pivotal role in the pathogenesis of infectious diseases, inflammatory disorders, allergic diseases, and autoimmune diseases in multiple organs, and external microenvironment imbalances as changes in the gut microbiota can affect or trigger the onset of a variety of diseases, such as immune disorders, metabolic diseases, and neurodegenerative disorders [4,5,6,7,8,9,10]. The complexity of the molecular relationship between intestinal dysbiosis and pathogenesis of immune disorders has prevented a complete understanding of the underlying processes [11,12].

Nonetheless, multiple reports have demonstrated that controlling intestinal immunity can help maintain systemic immunological homeostasis [5,11,12,13,14,15]. For example, studies in patients with inflammatory bowel disease, as well as in animal models, have shown benefit due to antibiotic treatment [5,16]. The pathogenesis of rheumatoid arthritis (RA) has been reported to be correlated with infectious microbes [17,18,19]. Animal models of RA show varied roles for microbiota in disease severity that range from inhibition to augmentation [17,18,19]. Additionally, antibiotic-mediated modification of the gut microbiota can significantly reduce disease severity or experimental autoimmune encephalomyelitis (EAE), a mouse model of multiple sclerosis [20,21]. In contrast, some commensals can also promote EAE development [22]. Therefore, gut microbiota strongly contributes to pathogenesis of autoimmunity at various steps during the disease process.

Non-obese diabetic (NOD) mice are an animal model of type 1 diabetes, as well as Sjögren’s syndrome-like lesion, wherein exocrine glands, such as lacrimal and salivary glands, are targeted by an autoimmune response [23,24]. Interestingly, the severity and incidence of insulitis and sialadenitis in NOD mice in a germ-free facility are significantly lower than that seen in a specific pathogen-free (SPF) facility [25]. Under SPF conditions, the incidence of diabetes varies by facility, suggesting that environmental changes affect gut microbiota, which can then influence autoimmune responses in distant organs, such as the pancreas and the salivary glands [26,27]. However, the precise cellular and molecular mechanisms underlying the relationship between the gut microbiota and the immune system remain unclear.

In this study, to understand whether the pathogenesis of autoimmunity involves changes in the intestinal microbiota, we evaluated the association between multiple autoimmune lesions in NOD mice and antibiotic treatment using immunopathological analysis. The findings of the current study may help comprehend novel pathogenic or regulatory mechanisms of autoimmunity that are affected by intestinal microbiota, apart from helping establish potential new treatments for autoimmunity.

## 2. Results

### 2.1. Antibiotic Treatment in NOD Mice

To change the composition of the intestinal microbiota in female NOD mice, we administered an antibiotic mixture comprising ampicillin, metronidazole, and neomycin in drinking water, which also contained an artificial sweetener, for 4 weeks (Figure 1a). Although a significant decrease in body weight was detected within 7 days of antibiotic treatment in the start in a group of mice administered with 0.33 g/L antibiotics, there was no difference in the body weight among the three groups after day 7 (Figure 1b). The quantity of 16S rDNA in the feces was used as readout for bacterial abundance, and antibiotic (0.33 g/L)-administered NOD mice showed a significantly lower copy number of 16S rDNA at day 24 compared to that at day 0 (Figure 1c). These results imply that antibiotic treatment in NOD mice led to a change in the composition of their intestinal microbiota.

### 2.2. Effect of Antibiotic Administration on the Immune System of NOD Mice

To evaluate the effect of antibiotic treatment on the immune system, we used flow cytometry to quantitate the proportion and numbers of immune cells in the spleen. Both proportion and number of CD8^+^ T cells and CD4^+^ T cells in the antibiotic-administered groups were similar to that of the control group (Figure 2a). Using two activation markers of T cells, namely, CD44 and CD62L, the effector memory phenotype (CD44^high^ CD62L^−^) of CD4^+^ T cells was evaluated, which also showed no change in the proportion and number of CD44^high^ CD62L^−^ CD4^+^ T cells due to administration of antibiotics (Figure 2b). Additionally, no changes were found in the proportion and numbers of Foxp3^+^ CD4^+^ regulatory T (Treg) cells (Figure 2c) or CD19^+^ B cells (Figure 2d) in the spleen. No changes were seen in immunoglobulin subclasses in the sera, including IgG1, IgG2c, and IgA, due to antibiotic administration (Figure 2e). These findings suggest that a change in the intestinal microbiota due to antibiotic administration did not affect the population or phenotype of T and B cells in NOD mice.

### 2.3. Effect of Antibiotics Administration on Cytokine Profile in NOD Mice

To evaluate changes in immune system of NOD mice administered with antibiotics, mRNA expression of cytokines in the spleen tissues was analyzed by quantitative RT-PCR. There were no differences in the T helper (Th)1 (IL-2 and TNF-α)-, Th2 (IL-4, IL-10, and TGF-β)-, or Th17 (IL-17)-type cytokines between control and antibiotic-treated NOD mice (Figure 3), suggesting that antibiotic administration did not influence cytokine profile of NOD mice in this study.

### 2.4. Effect of Antibiotic Administration on Autoimmune Sialadenitis in NOD Mice

Autoimmune lesions in female NOD mice are routinely observed in salivary glands [23,24], but histopathological analysis showed that inflammatory lesions with lymphocyte infiltration around the salivary duct were not influenced by antibiotic administration (Figure 4a). Further, there were no differences in focus score of the salivary gland lesion between control and antibiotic-administered mice (Figure 4b). Although the number of lymphocytes within the salivary gland tissue tended to be lower upon treatment with higher concentration of antibiotics (0.33 g/mL), there was no significant difference between control and antibiotic-administered mice (Figure 4b). Moreover, the numbers of CD4^+^ or CD8^+^ T cells in salivary gland tissues were no different between control and antibiotic-administered mice (Figure 4c), and there was no change in the mean fluorescence intensity of CD44 on CD4^+^ T cells due to antibiotic administration (Figure 4d). Therefore, changes in intestinal microbiota due to antibiotic administration for 4 weeks did not affect the development of autoimmune sialadenitis in NOD mice.

### 2.5. Effect of Antibiotic Administration on Autoimmune Insulitis in NOD Mice

We, next, evaluated the effect of antibiotic treatment on the onset and progression of type 1 diabetes or autoimmune insulitis measuring the concentration of blood glucose and by histopathological analysis of pancreas tissues. The concentration of blood glucose in antibiotic-administered NOD mice was similar to that of control mice (Figure 5a), and pathological analysis yielded no difference in insulitis or peri-insulitis in pancreatic tissues from control and antibiotic-administered NOD mice (Figure 5b). Additionally, there were no significant differences in pathological scores of peri-insulitis and insulitis between control and antibiotic-administrated mice (Figure 5c). These results demonstrate that intestinal microbial alteration by antibiotic administration does not influence the onset or progression of insulitis and type 1 diabetes.

## 3. Discussion

Here, we investigated whether autoimmune lesions in salivary glands and pancreatic islets in NOD mice are affected by antibiotic administration-induced changes in the enteral microenvironment. Even though antibiotic treatment significantly reduced bacterial DNA in the feces of NOD mice, pointing to a large imbalance in intestinal microbiota, these changes did not result in systemic effects on immune cell populations or functions, or the onset and progression of autoimmune lesions in these NOD mice.

A previous report has demonstrated that the autoimmune lesions, such as autoimmune sialadenitis and insulitis, are suppressed in NOD mice under germ-free conditions [25]. In addition, deletion of the toll-like receptor (TLR) adaptor protein, MyD88, in SPF NOD mice protects from them diabetes; in contrast, robust autoimmune diabetes is observed in germ-free *MyD88*−/− NOD mice [27]. Germ-free experiments using NOD mice focus on the relationship between the colonization of the intestinal environment by specific bacteria and development of autoimmune lesions. Whereas, our study addressed the effect of a large change in the composition of the intestinal microbiota. T and B cell populations, immunoglobulin production, and cytokine profile were not altered by antibiotic administration in our study. However, other studies in germ-free conditions have reported significant changes in various immune cell populations and functions, such as Tregs [28,29]. Nevertheless, in contrast to the results presented here, a previous report indicated that prolonged antibiotic administration promoted autoimmune diabetes in NOD mice [30], and in that report, vancomycin or neomycin, in the drinking water, was provided to pregnant mice just prior to birth, throughout lactation, and continued throughout the life span of the pups until diabetes onset [30]. Our study using antibiotics critically diverges from the protocol of the previous study in that we used a combination of antibiotics, shorter in duration of administration, and relatively provided a low dose of antibiotics. In fact, clinical administration of antibiotics to patients with infectious disease is thought to not induce any drastic change in the function of the immune system. Therefore, experiments with antibiotic administration may be clinically useful in understanding the effects of enteral microbiota changes on autoimmunity.

In relation to the effects on body weight, two factors are considered. One is body weight loss by any dysfunction of gastrointestinal tract, and the other is a decrease of food and water intake due to palatability of oral antibiotics [31]. To improve taste and palatability of the antibiotics, a high concentration (60 g/L) sweetener was added to the water bottles. After day 7, there was no difference between control and treated mice, suggesting that mice may get used to taking water with the taste and palatability of antibiotics, or gastrointestinal function may recover.

The effects of prebiotics in NOD mice, such as xylooligosaccharides and fecal microbiota transplants, includes alleviating autoimmune diabetes and sialadenitis [32]. Specific microbiota or molecules derived from certain bacteria are able to influence systemic immune cell populations or their functions to affect the onset or progress of autoimmunity. Several reports have analyzed the effects of microbiota on the progression of autoimmune lesions in EAE models and have described protective effects in germ-free mice, as either delayed onset of symptoms or complete protection from disease [21,22]. Antibiotic treatment in a mouse model of EAE has also been shown to prevent disease onset, further supporting a role for microbiota in EAE pathology [20]. Apart from ablation of the gut microbiota as a way to limit disease onset in EAE, administering bacteria with IL-10- and Treg-promoting effects, such as *Bacteroides fragilis*, is also efficacious in reducing the severity of EAE symptoms [33]. Additionally, the intestinal microbiota has been implicated in both induced and spontaneous mouse models of RA [17,18,19]. Specifically, in the spontaneous *IL-1rn*−/− mouse model of autoimmune arthritis, TLR4 signaling by the microbiome appears to be responsible for the induction of arthritis, as germ-free IL1rn−/− mice or conventional *IL-1rn*−/− *TLR4*−/− mice are protected from autoimmune arthritis [19]. Next, in germ-free, RA mouse models, monocolonization by Th17-inducing segmented filamentous bacteria rapidly induces the onset of autoimmune arthritis [17]. Similarly, alterations in the microbiota of human RA patients have also been widely reported with multiple studies linking an increase in *Prevotella* abundance in the gut with RA [18]. One study examining microbial differences between RA and healthy controls also found a significant association between *Prevotella copri* and susceptibility to arthritis [18]. In addition, a previous report demonstrated that a significantly different effect of antibiotic treatment was found on autoimmune lesions of female, but not male, NOD mice [34], suggesting that endocrine system controls critical immune responses, including infection and autoimmunity. In our study, antibiotics were orally administered to female NOD mice while ensuring immune system function, and antibiotic administration did not result in any change in specific immune cell populations. Immune system is maintained by many precise mechanisms, while many factors, including a drastic change of colonization of bacterial flora in the intestine after birth, induce dysfunction of immune cells. The dose or duration of antibiotic administration may also affect systemic immune response and autoimmune response. In relation to the specific microbial population of antibiotics-treated mice, we checked relative DNA abundance of *Lactobacillus* in the feces by q-PCR as a preliminary experiment. The relative DNA abundance of *Lactobacillus* was largely reduced by antibiotics treatment, suggesting that any specific microbial population may change by antibacterial effect (unpublished data). Although the relationship between autoimmunity and gut microbiota is still unclear in humans, various direct evidence regarding gut microbiota has emerged in inflammatory bowel disease, including ulcerative colitis and Crohn’s disease [35]. Multiple studies have implicated bacterial species and genes in the pathogenesis of IBD [35].

In summary, antibiotic administration did not affect the onset or progression of autoimmune lesions in NOD mice, and in the presence of a functional immune system, the autoimmune response appears to be independent of changes in enteral microbiota. These results are useful in determining the appropriate use of antibiotics in patients with autoimmune diseases.

## 4. Materials and Methods

### 4.1. Mice

NOD mice were bred and maintained in a SPF mouse colony in the animal facility at Tokushima University (Tokushima, Japan). Mice of five or less per cage were maintained under 12 h light /12 h dark cycle at 18 to 23 °C with 40 to 60% humidity condition, and received sterilized pellets by radiation (Oriental Yeast Co., LTD., Tokyo, Japan) and water ad libitum. This study was conducted according to the Fundamental Guidelines for Proper Conduct of Animal Experiment and Related Activities in Academic Research Institutions under the jurisdiction of the Ministry of Education, Culture, Sports, Science and Technology of Japan. The protocol was approved by the Committee on Animal Experiments of Tokushima University and Biological Safety Research Center, Japan (Permit Number: T29-115, March 6, 2017). All experiments were performed after administration of anesthesia, and all efforts were made to minimize suffering.

### 4.2. Antibiotics Administration

Female NOD mice (9−11 weeks old) were treated with a cocktail of broad-spectrum antibiotics 0.17 or 0.33 g/L ampicillin (FUJIFILM Wako Pure Chemical Corp., Osaka, Japan), 0.17 or 0.33 g/L metronidazole (FUJIFILM Wako Pure Chemical Corp.), and 0.17 or 0.33 g/L neomycin (Thermo Fisher Scientific, Waltham, MA, USA) dissolved in drinking water with 60 g/L artificial sweetener (MENASHA Corp., Neenah, WI, USA). We conducted a preliminary experiment to determine the concentration of the antibiotics in our study according to the information of the several previous reports [36,37]. When 0.17, 0.33 or 1 mg/mL antibiotics with a sweetener were administered for mice, some mice treated with 1.0 mg/mL antibiotics died within 2 weeks after start. Therefore, we determined the concentration (0, 0.17, and 0.33 mg/mL) in our experiments. Drinking water with antibiotics in a shaded bottle was changed every 3−4 days. Mice received the mixture water for 4 weeks and they were euthanized at 13−15 weeks of age, as applicable. Mice of the control group received the mixture water without antibiotics.

### 4.3. Cell Isolation

For the isolation of immune cells from the salivary gland, a lateral salivary gland lobe was minced into 1–3 mm pieces and was digested with collagenase (1 mg/mL, FUJIFILM Wako Pure Chemical Corp.), hyaluronidase (1 mg/mL, Sigma-Aldrich Co. St. Louis, MO, USA), and DNase (10 μg/mL, Roche Diagnostics K.K., Tokyo, Japan) in Dulbecco’s modified Eagle’s medium (DMEM) containing 10% fetal bovine serum (FBS) at 37 °C for 40 min using the gentleMACS Dissociator (Miltenyi Biotec, Bergisch Gladbach, Germany). Next, splenocytes were homogenized in DMEM containing 2% FBS using a gentleMACS Dissociator (Miltenyi Biotec). Red blood cells were removed from the spleen cells using 0.83% ammonium chloride, and the viability and number of the isolated cells were evaluated on a Luna II cell counter (Logos Biosystems, Gyeonggi-do, Korea) using trypan blue staining. Subsequently, a proportion of the suspended cells was analyzed by flow cytometry. The absolute number of immune cells was calculated using data on total cell number and proportion of each cell type. As for the salivary gland, we used lateral lobe to determine the cell number and the proportion of immune cells.

### 4.4. Flow Cytometric Analysis

Immune cells were stained using antibodies against FITC-conjugated anti-mouse CD19 (ID3, TONBO Bioscience, San Diego, CA, USA), CD44 (IM7, eBioscience, San Diego, CA, USA), and Foxp3 (FJK-16s, eBioscience), PE-conjugated anti-mouse CD8 (53-6.7, eBioscience), PE-Cy7-conjugated anti-mouse CD4 (GK1.5, TONBO Bioscience), APC-conjugated anti-mouse CD62L (MEL-14, Invitrogen, Carlsbad, CA, USA), and APC-Cy7-conjugated anti-mouse CD45.2 (A20, BioLegend). A FACSCanto flow cytometer (BD Biosciences, San Jose, CA, USA) was used to quantify cell populations based on expression profile. Isotype antibodies were used for analysis of each surface molecule as a control. Immune cells obtained from spleen and salivary gland tissues in NOD mice were analyzed according to a gating strategy as previously described [38]. Briefly, lymphocyte subset was checked by gating on side scatter (SSC)/forward scatter (FSC). Then, single cells were obtained by discriminating doublets using FSC-Hight (H)/FSC-Area (A), and viable immune cells were gated on CD45.1^+^ and 7-aminoactinomycin D (7AAD)^−^ (Invitrogen). Data were analyzed using the FlowJo FACS Analysis software (BD Biosciences).

### 4.5. Quantitative Reverse Transcription-PCR

Total RNA from the spleen was extracted using the RNAiso Plus kit (TakaRa Bio Inc., Shiga, Japan) according to the manufacturer’s instructions. Total RNA was then reverse-transcribed into cDNA using the PrimeScript RT Master Mix (Takara Bio Inc.). The transcripts of target genes and β-actin from the spleen were generated on a Light Cycler 96 System (Roche) using TB Green Premix Ex Taq II (Takara Bio Inc.) and the following primers: IL-2: forward, 5′-CCTGAGCAGGATGGAGAATTACA-3′ and reverse, 5′-TCCAGAACATGCCGCAGAG-3′; IL-4: forward, 5′-TCTCATGGAGCTGCAGAGACTCT-3′ and reverse, 5′-TCCAGGAAGTCTTTCAGTGATGTG-3′; IL-10: forward, 5′-ATCGATTTCTCCCCTGTGAA-3′ and reverse, 5′- TGTCAAATTCATGGCCT-3′; IL-17: forward, 5′-AGTGTTTCCTCTACCCAGCAC-3′ and reverse, 5′-GAAACCGCCACCGCTTAC-3′; TNF-α: forward, 5′-CCTCCTGGCCAACGGCATG-3′, and reverse, 5′-CTCCACTTGGTGGTTTGCTA-3′; TGF-β: forward, 5′-GACCGCAACAACGCCATCTAT-3′, and reverse 5′-GGCGTATCAGTGGGGGTCAG-3′; β-actin: forward, 5′-GACGGCCAGGTCATCACTAT-3′, and reverse 5′-CTTCTGCATCCTGTCAGCAA-3′. Relative mRNA expression of each transcript was normalized against β-actin mRNA.

### 4.6. Detection of Bacterial DNA

Feces were collected at 10 h after cleaning the cage. Fecal DNA was extracted using the ISOSPIN Fecal DNA kit (NIPPON GENE, Tokyo, Japan) according to the manufacturer’s instructions. To quantify the bacterial density of feces, real-time PCR was performed using primers targeting the conserved region of the bacterial 16S rDNA gene. A standard curve was prepared by serial dilution of the plasmid containing one copy of the 16S rDNA amplicon (qCR-Blunt II-TOPO, Invitrogen) [39]. The primer sets were used as followed; 16S rDNA: forward, 5′-ACTCCTACGGGAGGCAGCAGT-3′, and reverse, 5′-ATTACCGCGGCTGCTGGC-3′.

### 4.7. Histological Analysis

Salivary gland and pancreas tissues were fixed with 10% phosphate-buffered formaldehyde (pH 7.2) and prepared for histological examination. Sections were stained with hematoxylin and eosin (H&E). Focus score in submandibular gland tissues was determined by counting foci; foci were defined as lesions with 50 or more tightly aggregated lymphocytes within a 4 mm^2^ area in the salivary gland tissue. In addition, the number of lymphocytes infiltrating in the entire tissue section were counted per mm^2^ area. For semiquantitative evaluation of pancreatic infiltration, histological analysis of sections containing the islets was performed. In brief, the degree of cellular infiltration was scored from 0 to 5 as follows: 0 = no inflammation; 1 = infiltrates in small foci at the islet periphery; 2 = infiltrates surrounding the islets (peri-insulitis); 3 = intra-islet infiltration <50% of the islet but islet derangement; 4 = extensive infiltration in 50% of the islet, cell destruction, and prominent cytoarchitectural derangement; 5 = islet atrophy because of β-cell loss.

### 4.8. Enzyme-Linked Immunosorbent Assay

Serum IgG1, IgG2c and IgA were measured using mouse ELISA kits (Bethyl Laboratories, Montgomery, TX, USA) according to the manufacturer’s instructions. Non-specific sites were absorbed using Blocking One (NACALAI TESQUE, INC., Kyoto, Japan). The OD at 490 nm for each well was measured on a SpectraMax i3 microplate reader (Molecular Devices, San Jose, CA, USA).

### 4.9. Blood Sugar Level Measuring

Blood glucose levels were measured in 10 μL of venous blood using CareFast R (NIPRO, Osaka, Japan).

### 4.10. Statistical Analysis

The differences between individual groups were determined using two-tailed Student’s *t*-test, and *p* < 0.05 was considered statistically significant. Data are presented as mean ± standard deviation (SD)

## 5. Conclusions

The onset or development of autoimmunity appears to be independent of antibiotic treatment-mediated changes in the gastrointestinal environment in the presence of a functional immune system.

## Figures and Tables

**Figure 1 ijms-22-03239-f001:**
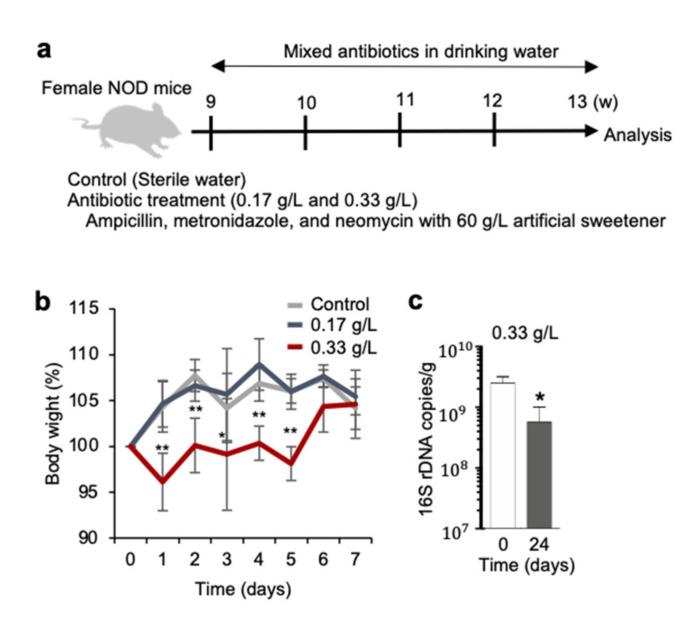
Administration of antibiotics to non-obese diabetic (NOD) mice. (**a**) Antibiotics (0.17 and 0.33 g/L) were mixed in drinking water along with an artificial sweetener (60 g/L) and were administered to NOD mice during 4 weeks from 9 to 13 weeks of age. (**b**) Body weight was monitored during antibiotic administration. Relative change in body weight was determined by comparing body weight at experiment start and end. All experiments were repeated four times. Data are presented as mean ± SD of 16 mice in control and 0.33 g/L groups, and for 9 mice in the 0.17 g/L group; * *p* < 0.05, ** *p* < 0.01. (**c**) Bacterial 16S rDNA was determined by quantitative PCR. Data are presented as mean ± SD of three mice in each group; * *p* < 0.05.

**Figure 2 ijms-22-03239-f002:**
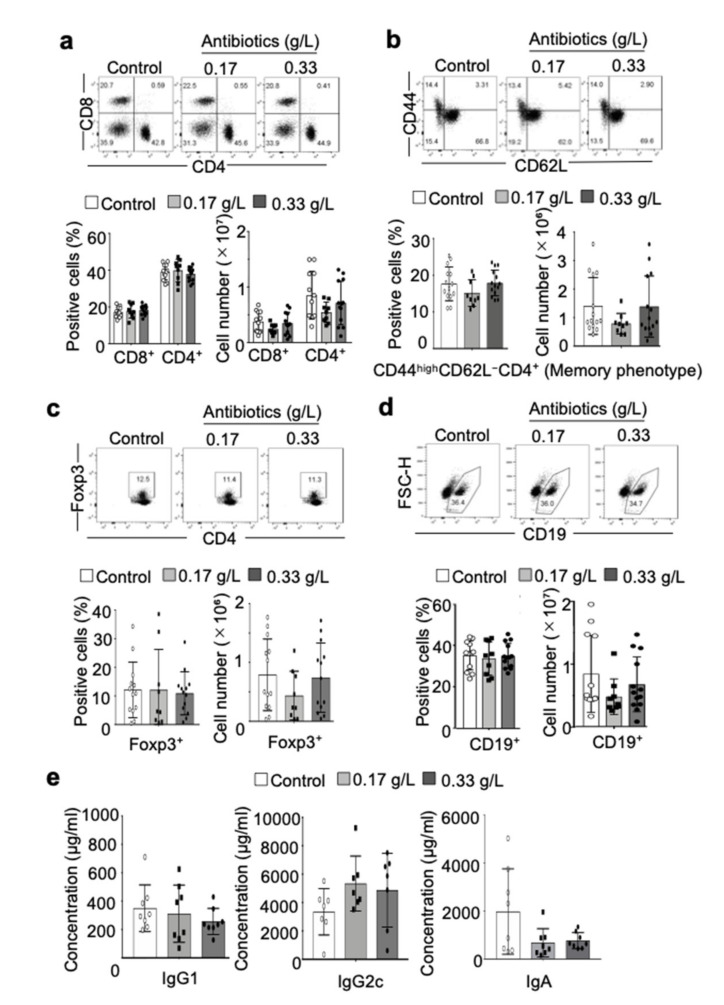
Effect of antibacterial treatment on immune cells. (**a**) T cell populations in the spleen were evaluated and quantified by flow cytometric analysis. Proportion and number of CD4^+^ or CD8^+^ T cells were calculated, and data are presented as mean ± SD of 13 mice in control and 0.33 g/L group, 9 mice in 0.17 g/L group. (**b**) Memory phenotype of CD4^+^ T cells in the spleen was assessed by flow cytometric analysis using anti-CD44 and CD62L antibodies. Proportion and number of CD44^high^ CD62L^−^ CD4^+^ T cells were calculated, and data are presented as mean ± SD of 13 mice in control and 0.33 g/L groups and 9 mice in the 0.17 g/L group. (**c**) Regulatory T (Treg) cells in spleen were quantified by flow cytometric analysis using intracellular staining of anti-Foxp3 antibody. Proportion and number of Foxp3^+^ CD4^+^ T cells were calculated, and data are presented as mean ± SD of 13 mice in control and 0.33 g/L groups, and 9 mice in the 0.17 g/L group. (**d**) B cells of spleen were evaluated and quantified by flow cytometric analysis. Proportion and number of CD19^+^ B cells were calculated, and data are presented as mean ± SD of 13 mice in control and 0.33 g/L group, and 9 mice in the 0.17 g/L group. (**e**) Serum IgG1, IgG2c, and IgA were measured by ELISA. Concentration levels of IgG1, and IgA were calculated, and data are presented as mean ± SD of seven or eight mice in each group.

**Figure 3 ijms-22-03239-f003:**
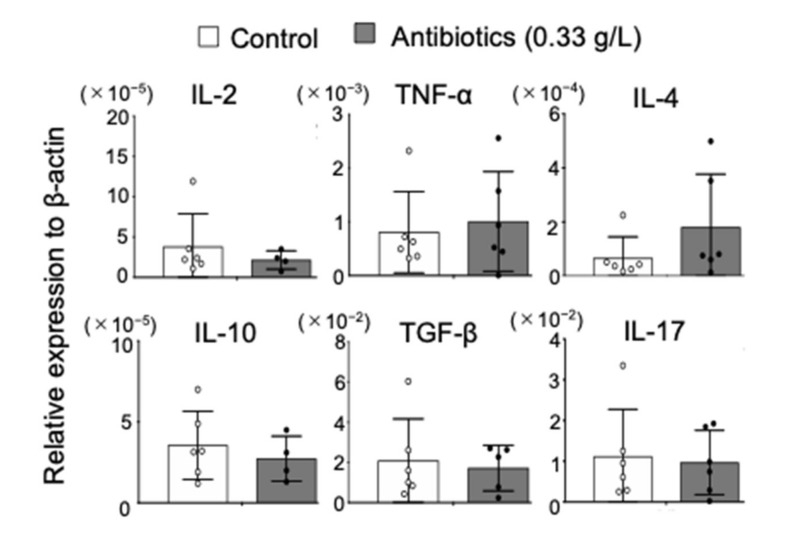
Effect of antibiotic administration on cytokine mRNA expression. Th1 (IL-2 and TNF-α)-, Th2 (IL-4, IL-10, and TGF-β)-, and Th17 (IL-17)-type cytokines in the spleen of control and antibiotic (0.33 g/L)-administered NOD mice were evaluated by quantitative Reverse Transcription-PCR (RT-PCR). Data are presented as mean ± SD of four to six mice in each group.

**Figure 4 ijms-22-03239-f004:**
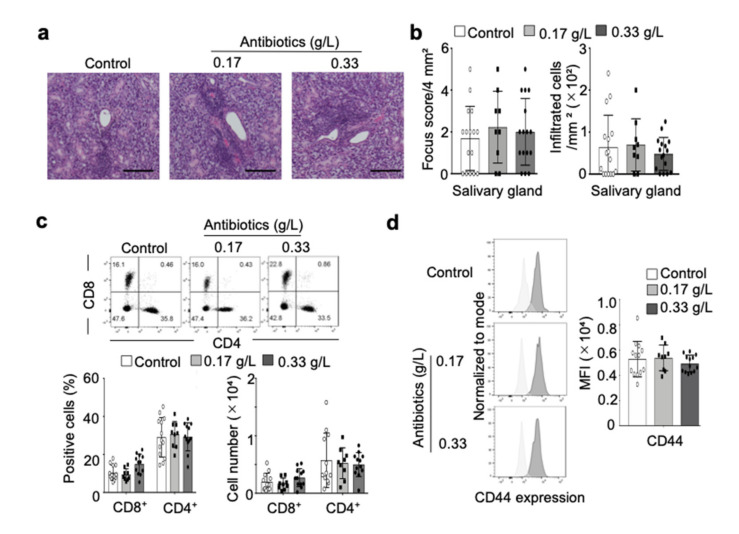
Effect of antibiotic administration on autoimmune lesions in salivary gland tissues of NOD mice. (**a**) Pathological images of salivary gland tissues were shown. Images are of representative mice in each group. Scale bar: 100 μm. (**b**) Focus score per 4 mm^2^ was assessed using pathological sections. Number of infiltrated cells within salivary gland tissues (1 mm^2^) were histologically semi-quantified, and data are presented as mean ± SD of 16 mice in control and 0.33 g/L groups and 9 mice in the 0.17 g/L group. (**c**) T cell populations in salivary gland tissues were detected by flow cytometric analysis. Proportion and number of CD4^+^ or CD8^+^ T cells were calculated, and data are presented as mean ± SD of 13 mice in control and 0.33 g/L groups, and 9 mice in the 0.17 g/L group. (**d**) CD44 expression on CD4^+^ T cells in salivary gland tissues was measured by flow cytometric analysis. Data are presented as mean fluorescence intensity of CD44 expression ± SD of 13 mice in control and 0.33 g/L groups and 9 mice in the 0.17 g/L group.

**Figure 5 ijms-22-03239-f005:**
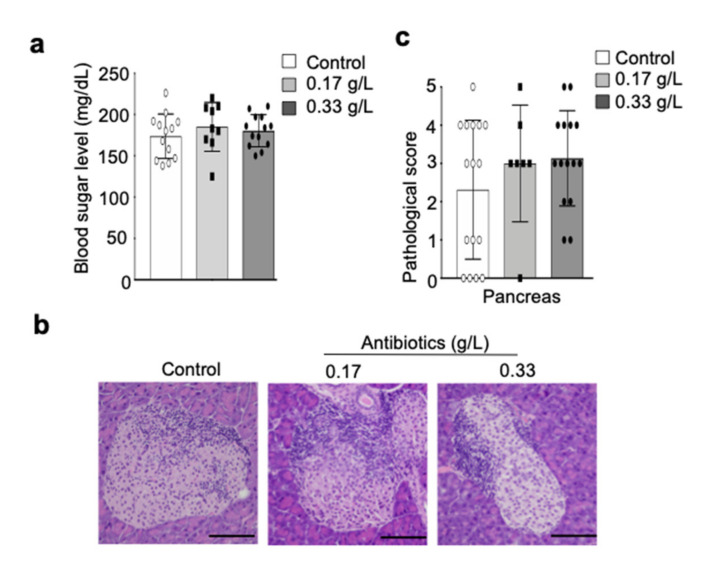
Effect of antibiotic administration on autoimmune diabetes in NOD mice. (**a**) Blood glucose concentration in control and antibiotic-administered (0.17 and 0.33 g/L) mice were determined by blood glucose meter. Data are presented as mean ± SD of 13 mice in control and 0.33 g/L groups and 9 mice in the 0.17 g/L group. (**b**) Pathological photos of pancreas tissue sections stained with H&E are shown, and the results are representative of 16 mice in control group and 7 mice in 0.17 g/L group, and 15 mice in 0.33 g/L group. Scale bar: 100 μm. (**c**) Pathological score of pancreas tissues was evaluated, and data are presented as mean pathological score ± SD of 16 mice in control group, 7 mice in the 0.17 g/L group, and 15 mice in the 0.33 g/L group.

## Data Availability

Data that support the results of the present study are available from the corresponding author upon reasonable request.

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
