# Peer review of "Formation of Autoimmune Lesions Is Independent of Antibiotic Treatment in NOD Mice"

_ijms, 2021, doi:10.3390/ijms22063239_

Round 1

Reviewer 1 Report

Overall, the manuscript is well written and the data are clearly presented. The objective of the study is to investigate the potential role of gut microbiome on the onset and progression of autoimmune responses in NOD mice.

The relevance of the NOD animal model for studying Sjogren’s has been extensively discussed elsewhere. This model, amongst other signs, exhibits only a few signs of Sjogren’s pathology including sialadenitis and is far from been named Sjogren’s model. The current knowledge suggests that NOD is a model of Type I diabetes and sieladenitis, and thus, in lines 43-44 a correction is needed about it.

Since there is no significant difference in any of the parameters studied between the two groups, there are not many things that can be questioned. The immune profiling of spleen cells and salivary glands is adequate and very informative about the effects of the microbiome. The experiments are clear and enough to support the central “no change” finding.

A few things that need to be clarified:

  1. How were the concentrations of different antibiotics chosen?
  2. Lines 63-66: How can the body weight gain difference be explained? Was there a difference in food intake by day 7 that could explain that?
  3. Can the authors discuss the specific microbial populations that were affected by the antibiotics and the relevance with human gut micrbiome?
  4. In flow cytometry experiments, what were the control antibodies? The authors should mention in the Methods part the complete flow cytometry protocol including unstained controls, FMO, isotype controls etc. as well as the gating strategy to obtain singlets for further analysis in flowjo.

Author Response

To Reviewers,

Thank you so much for evaluating and reviewing our manuscript. We modified the manuscript and figures according to the valuable suggestions and comments by the reviewers.

Rev #1,

The relevance of the NOD animal model for studying Sjogren’s has been extensively discussed elsewhere. This model, amongst other signs, exhibits only a few signs of Sjogren’s pathology including sialadenitis and is far from been named Sjogren’s model. The current knowledge suggests that NOD is a model of Type I diabetes and sialadenitis, and thus, in lines 43-44 a correction is needed about it.

Answer: As pointed out by Reviewer #1, the lesions of salivary and lacrimal glands in NOD mice exhibit relatively mild inflammation unlike those the other SS model and the patients. Therefore, we modified Sjögren’s syndrome to Sjögren’s syndrome-like lesion in the Introduction Section of the revised manuscript.

A few things that need to be clarified:

  1. How were the concentrations of different antibiotics chosen?

Answer: We determined the concentration of antibiotics in this study according to the information of the several previous reports. The description was added to the Materials and Methods Section of the revised manuscript al below:

We conducted a preliminary experiment to determine the concentration of the antibiotics in our study according to the information of the several previous reports [35,36]. When 0.17, 0.33 or 1 mg/ml antibiotics with a sweetener were administered for mice, some mice treated with 1.0 mg/ml antibiotics died within 2 weeks after start. Therefore, we determined the concentration (0, 0.17, and 0.33 mg/ml) in our experiments.

The description was added to the Results Section of the revised manuscript.

  1. Lines 63-66: How can the body weight gain difference be explained? Was there a difference in food intake by day 7 that could explain that?

Answer: Regarding the effect on body weight, two factors are considered. One is body weight loss by any dysfunction of gastrointestinal tract, and the other is a decrease of food and water intake due to palatability of oral antibiotics [30]. To improve taste and palatability of the antibiotics, a high concentration (60 g/L) sweetener was added to the water bottles. After day 7, there was no difference between control and treated mice, suggesting that mice may get used to taking water with the taste and palatability of antibiotics, or gastrointestinal function may recover. The description was added to the Discussion section of the revised manuscript.

  1. Can the authors discuss the specific microbial populations that were affected by the antibiotics and the relevance with human gut micrbiome?

Answer: Regarding to the specific microbial population of antibiotics-treated mice, we checked relative DNA abundance of Lactobacillus in the feces by q-PCR as a preliminary experiment. The relative DNA abundance of Lactobacillus was largely reduced by antibiotics treatment, suggesting that any specific microbial population may change by antibacterial effect (unpublished data).

Although relationship between autoimmunity and gut microbiota is still unclear in human, many direct evidences regarding gut microbiota are emerging in inflammatory bowel disease, including ulcerative colitis and Crohn’s disease [34]. Multiple studies have implicated bacterial species and genes in the pathogenesis of IBD [34].

The description was added to the Discussion Section of the revised manuscript.

  1. In flow cytometry experiments, what were the control antibodies? The authors should mention in the Methods part the complete flow cytometry protocol including unstained controls, FMO, isotype controls etc. as well as the gating strategy to obtain singlets for further analysis in flowjo.

Answer: The control and gating strategy in flow cytometric analysis was added in detail in the Materials and Methods Section of the revised manuscript as below:

Isotype antibodies were used for analysis of each surface molecule as a control.

Immune cells obtained from spleen and salivary gland tissues in NOD mice were analyzed according to a gating strategy as previously described [37]. Briefly, lymphocyte subset was checked by gating on side scatter (SSC)/forward scatter (FSC). Then, single cells were obtained by discriminating doublets using FSC-Hight (H)/FSC-Area (A), and viable immune cells were gated on CD45.1+ and 7-aminoactinomycin D (7AAD) (Invitrogen). Data were analyzed using the FlowJo FACS Analysis software (BD Biosciences).

Reviewer 2 Report

This is an interesting study. I just have minor concerns.

  1. In the abstract, "These results suggest that the onset and progression of autoimmunity is independent of enteral microbiota changes" is not accurate. The enteral microbiota changes by the antibiotics treatment in this study may not reflect the key microbial changes in autoimmunity. I suggest to change to "These results suggest that the onset and progression of autoimmunity may be independent of enteral microbiota changes".

  2. The significance was not clearly labeled in some figures such as figure 2
  3. Some figures are not very clear. The quality needs to be improved.

Author Response

To Reviewers,

Thank you so much for evaluating and reviewing our manuscript. We modified the manuscript and figures according to the valuable suggestions and comments by the reviewers.

Rev#2

  1. In the abstract, "These results suggest that the onset and progression of autoimmunity is independent of enteral microbiota changes" is not accurate. The enteral microbiota changes by the antibiotics treatment in this study may not reflect the key microbial changes in autoimmunity. I suggest to change to "These results suggest that the onset and progression of autoimmunity may be independent of enteral microbiota changes".

Answer: According to the suggestion by Reviewer #2, the description was re-written in the revised manuscript.

  1. The significance was not clearly labeled in some figures such as figure 2

Answer: The quality of images including a significant symbol was improved in the new Figure 2.

  1. Some figures are not very clear. The quality needs to be improved.

Answer: The quality of images was more improved in the new figures.